# Numerical Modeling of Vortex-Based Superconducting Memory Cells: Dynamics and Geometrical Optimization

**DOI:** 10.3390/nano14201634

**Published:** 2024-10-12

**Authors:** Aiste Skog, Razmik A. Hovhannisyan, Vladimir M. Krasnov

**Affiliations:** Department of Physics, Stockholm University, AlbaNova University Center, SE-10691 Stockholm, Sweden; aisk1747@student.su.se (A.S.);

**Keywords:** Josephson effect, superconductivity, digital electronics, nanodevices

## Abstract

The lack of dense random-access memory is one of the main obstacles to the development of digital superconducting computers. It has been suggested that AVRAM cells, based on the storage of a single Abrikosov vortex—the smallest quantized object in superconductors—can enable drastic miniaturization to the nanometer scale. In this work, we present the numerical modeling of such cells using time-dependent Ginzburg–Landau equations. The cell represents a fluxonic quantum dot containing a small superconducting island, an asymmetric notch for the vortex entrance, a guiding track, and a vortex trap. We determine the optimal geometrical parameters for operation at zero magnetic field and the conditions for controllable vortex manipulation by short current pulses. We report ultrafast vortex motion with velocities more than an order of magnitude faster than those expected for macroscopic superconductors. This phenomenon is attributed to strong interactions with the edges of a mesoscopic island, combined with the nonlinear reduction of flux-flow viscosity due to the nonequilibrium effects in the track. Our results show that such cells can be scaled down to sizes comparable to the London penetration depth, ∼100 nm, and can enable ultrafast switching on the picosecond scale with ultralow energy per operation, ∼$10^{-19}$ J.

## 1. Introduction

During the era of vacuum tube electronics, two technologies emerged as potential new paradigms of digital computing. The transistor was invented in 1947, setting off semiconductor electronics [1]. A decade later, the development of the first superconductive switching device, the cryotron [2], marked the beginning of superconductive electronics. Competition between the two ended in favor of semiconductors in 1983 [3,4].

Nowadays, entire industries and critical infrastructures depend on data centers housing huge server farms. The amount of processed data has been growing exponentially with time [5], accompanied by the increasing computer system complexity [6] and explosive power consumption, which could become a bottleneck for further development [7]. Simultaneously, physical limits of semiconductor electronics are being approached [8]. Gate leakage makes it difficult to make transistors smaller and the RC time constant limits the operation speed. It has been argued that the end of Moore’s law [9], describing the development of semiconductor electronics for a long time, has already been reached [10]. Therefore, a paradigm shift in high-end computation is needed yet again.

Superconducting electronics could enable a dramatic increase of the calculation speed and power efficiency [11]. However, the existing RSFQ (rapid single-flux quantum) electronics has a major problem with scalability and does not allow very large-scale integration (VLSI) needed for high-end computation. The lack of dense random access memory (RAM) is considered as one of the main bottlenecks for building a digital superconducting computer [11,12,13,14]. Several non-RSFQ approaches for superconducting RAM have been proposed [15,16,17,18,19,20,21,22,23,24,25,26] and are awaiting experimental scrutiny.

In Refs. [23,24] it was suggested to employ a single Abrikosov vortex (AV) as an information carrier in digital superconducting electronics. An AV is the smallest quantized magnetic object in superconductors with the size determined by the London penetration depth, $\lambda_{L}$∼100 nm. This allows miniaturization to sub-micron sizes for operation at zero magnetic field [24]. Vortices can be manipulated by current [23,24,26,27,28], magnetic field [29,30], heat [27,31] and light [31] pulses. The AVRAM cell essentially represents a fluxonic quantum dot [32,33,34,35,36] in which vortex interaction with the edges and quantum confinment effects are strong and where cell geometry, sizes of components and spatial asymmetry are playing important roles. Therefore, conscious geometrical engineering is required for the optimization of such devices [24].

In this work, we perform the systematic numerical modeling of AVRAM cell dynamics at zero external magnetic field using time-dependent Ginzburg–Landau (TDGL) equations. The cell consists of a mesoscopic-sized thin-film superconducting island with a vortex trap, a notch for vortex entry placed asymmetrically on one edge of the film and a track with reduced-order parameters for guiding a vortex to the trap. We are aiming for the optimization of geometrical parameters and clarification of the cell’s performance and limitations. We observe that optimal sizes of cell components are determined by the London penetration depth. Conditions for controllable write and erase operations with short current pulses are determined. Most remarkably, we observe ultrafast cell switching with vortex velocities more than an order of magnitude faster than expected for flux-flow motion in a uniform macroscopic superconductor. This phenomenon is attributed to a combination of strong nonlinear reduction of viscosity caused by enhanced nonequilibrium effects in the track and to a strong interaction of AVs with the edges of the mesoscopic island. We conclude that AVRAM cells, operational at zero applied field, can be scaled down to sizes comparable to $\lambda_{L}$∼100 nm, and they can facilitate ultrafast, ∼ps, switching with ultralow, ∼0.1 aJ, energy per operation. Therefore, AVRAM cells exemplify a VLSI-compatible and energy-efficient alternative for superconducting memory.

## 2. Results

The digital 0 and 1 states in AVRAM correspond to the states without and with a trapped AV. The operation of AVRAM cells requires a deterministic control of vortex introduction into (write) and removal from (erase) the cell by supplying short current pulses [23,24,26,28]. Controllability is the key challenge. As discussed in Ref. [24], it necessitates a conscious geometrical design of the cell. Figure 1 shows a scanning electron microscope (SEM) image of an AVRAM cell prototype from Ref. [24]. It contains a mesoscopic Nb island with sizes 1 × 1 μm^2^, comparable to $\lambda_{L}$, a vortex trap (hole), two readout Josephson junctions (JJs) and a guiding track placed asymmetrically on one side of the trap.

Figure 2a shows the cell geometry considered in this work. It consists of a rectangular superconducting island with dimensions $L_{x}=1\;$μm and $L_{z}=2$ μm, containing a circular vortex trap in the middle—a symmetric hole at x=z=0 with a diameter *D*, a half-oval notch on the right edge of the film and a vortex-guiding track with reduced superconductivity and the width *W*, connecting the notch and the trap. Compared to the cell prototype from Figure 1, it is rotated by 90 degrees so that the right edge of the simulated cell corresponds to the test JJ at the bottom.

Numerical modeling is performed by solving the TDGL equation using the pyTDGL package [37]. We assume the following parameters: $\lambda_{L}=179$ nm, the coherence length, ξ=44 nm, the TDGL relaxation time, $\tau_{GL}=0.27$ ps and the normal conductivity $\sigma_{n}=6.57$ $10^{4}$ (${\Omega \mathrm{cm})}^{-1}$. The simulations are purely two-dimensional, and there are no stray fields in free space. Therefore, Pearl length [38] does not play a role, and the current flow range is determined by $\lambda_{L}$. For the same reason, the film thickness, *d*, does not play any role, other than scaling of the total current. Therefore, we will normalize the current to the Ginzburg–Landau depairing current, $I_{dp}$. A more detailed description of the simulations is provided in Appendix A. All presented calculations are carried out at zero applied magnetic field.

Vortex manipulation is achieved by sending current pulses through the terminals placed at the bottom and the top edges of the film. Since the magnetic field of the vortex in a thin film is directed perpendicularly to the film, such pulses induce a Lorentz force in the horizontal (left–right) direction,
(1)$$F_{L}=d\left[J\times \Phi_{0}\right],$$
where $\Phi_{0}$ is the flux quantum and *J* is the current density.

Several time scales are important for AVRAM operation: the 0→1 write time, $\tau_{01}$, is the time it takes for the vortex to arrive in the previously empty trap after turning on the current; the 1→0 erase time, $\tau_{10}$, is the time for the initially trapped AV to leave the device after turning on the current in the opposite direction; and the residence time, $\tau_{r}$, is the time period during which the vortex remains in the trap while current is applied. The device is reciprocal with respect to direction of the current flow. Positive current writes a vortex which is erased with a negative current, and negative current writes an antivortex erased with a positive current. Equal magnitudes of the corresponding currents would result in equal write and erase times for a vortex or an antivortex. We will consider the former case—writing a vortex and erasing it with negative current.

### 2.1. The Role of Spatially Asymmetric Track and Notch

As emphasized in Ref. [24], spatially asymmetric geometry is necessary for controllable vortex manipulation. In our case, the asymmetry is introduced by the notch and the track, which is placed on the right side of the cell. Without them, in a perfectly symmetric cell, it was not possible to introduce vortices at all. The film stayed in the Meissner state almost up to $I_{dp}$, above which the whole film turned into the resistive state. Adding a symmetric track spanning the entire horizontal length of the film allows for the introduction of vortices, but it does not allow vortex trapping. In this case, a vortex and an antivortex enter simultaneously from the opposing sides of the film, move toward each other and annihilate without being trapped [39,40].

An asymmetric notch enables a controllable entrance of the vortex from one edge of the cell [41]. The notch creates an asymmetric distribution of supercurrent with enhanced current density near the tip of the notch, see Figure 3d, which becomes the preferable entrance point for the AV. Thus, both the notch and the track are crucial for vortex manipulation.

In what follows, we will consider a fixed semi-oval notch geometry (110×110 nm^2^). Guiding the vortex into the trap is the easiest if the notch is aligned with the trap, $z_{n}=0$; then, the traversal path is the shortest and coincidental with the direction of the Lorentz force. However, generally, the notch could be shifted by some distance $z_{n}$, as sketched in Figure 2a. In this case, if the tack is not “deep” enough, the traversing vortex may diverge from it [40,42] and miss the trap. The “depth” is defined in terms of the reduced order parameter, ${\left|\psi \right|}^{2}$, in the track, and its width, *W*. Below, we will show results for a moderately deep track with ${\left|\psi \right|}^{2}=0.6$ compared to the equilibrium value in the film.

Figure 2b shows the 0→1 write time as a function of misalignment, $z_{n}$, between the notch and the trap. Calculations are made for $I/I_{dp}=0.78$, the trap diameter, $D≃\lambda_{L}$, and the track width, $W≃0.97\;\lambda_{L}$. The fastest trapping occurs in the aligned case, $z_{n}=0$. With increasing misalignment, the trapping time increases approximately parabolically, and it is qualitatively consistent with $\tau \propto x^{2}+z^{2}$, which is expected for the viscous vortex motion along the track.

Figure 2c shows $\tau_{01}$ as a function of the track width for $z_{n}=-500$ nm and other parameters the same as in Figure 2b. Too narrow tracks, W<58 nm∼ξ, were not able to guide the vortex all the way to the trap. For $\xi <W<\lambda_{L}$, $\tau_{01}$ rapidly decreases with increasing *W*, saturating for wider tracks, W≃200 nm $>\lambda_{L}$. Analysis of the trap-size, dependence, $\tau_{01}\left(D\right)$, revealed a qualitatively similar behavior; see Figure 4c below. Therefore, we conclude that the optimal track width and trap diameter are determined by the London penetration depth, $D≃W≃\lambda_{L}$.

### 2.2. Vortex Velocity in the Mesoscopic Limit: Edge Interaction and Non-Equilibrium Effects

RAM should have short write/erase times. For the target clock frequency of 100 GHz [43], it should be below 10 ps. For vortex-based electronics, the switching time is determined by the vortex time of flight,
(2)$$\tau =L_{x}/2v,$$
where $L_{x}/2$ is the distance from the edge to the trap and *v*—vortex velocity. Smaller devices with faster vortices are needed for the high-frequency operation.

Although an AV is a practically massless electromagnetic object, it usually propagates at velocities much slower that the speed of light [44,45,46,47] (unlike Josephson vortices in tunnel JJs). This is caused by a large viscous damping due to power dissipation in the normal core. According to the Bardeen–Stephen model [48], the viscosity is
(3)$$\eta ≃\frac{\Phi_{0}^{2}\sigma_{n}d}{2\pi \xi^{2}}.$$
Together with Equation (Equation 1), it predicts a linear growth of v(J),
(4)$$v≃\frac{2\pi \xi^{2}}{\Phi_{0}\sigma_{n}}J.$$
However, this linear approximation is valid only at low velocities *v*∼1km/s [45,46] corresponding to small *J* compared to the GL depairing current density,
(5)$$J_{dp}=\frac{\Phi_{0}}{3\sqrt{3}\mu_{0}\pi \lambda_{L}^{2}\xi }.$$

Our aim is to understand the dynamics of a single vortex in mesoscopic devices with *L*∼$\lambda_{L}$. Such fluxonic quantum dots are characterized by a strong interaction between the AV and the device edges [32,33,34,35,36,40]. To understand the role of edge effects for flux flow in the mesoscopic limit, we first consider a cell without a trap, as shown in Figure 5a. When the current is applied, the AV enters through the notch at the right edge, moves along the track and leaves the device through the left edge. Figure 5b,c represent the snapshot and the cross-section of the order parameter when the vortex passes the middle of the island.

Figure 5d shows AV velocities along the track at three applied currents, $I/I_{d}=0.31$ (blue), 0.41 (red) and 0.49 (green). We show mean velocity values in four sections of the track, $x_{4}=\left[-500,-365.8\right]$ nm, $x_{3}=\left[-365.8,-190\right]$ nm, $x_{2}=\left[-190,0\right]$ nm, $x_{1}=\left[0,255.8\right]$ nm, and $\langle v_{i}\rangle =\Delta x_{i}/\Delta t_{i}$, where $\Delta t_{i}$ is the AV time of flight through the section. Remarkably, we observe a rapid increase of vortex velocity along its trajectory despite an approximately constant Lorentz force. Figure 5e shows the current dependence of mean velocities in the four sections.

Since an AV is practically massless, it always moves at an equilibrium: viscosity-limited velocity. Therefore, the increase of vortex velocity is not due to acceleration (inertia) but is rather a consequence of additional spatially inhomogenous edge forces. Edge effects can be described as the interaction of a vortex with an array of image vortices and antivortices [49]. Generally, this leads to an attractive force between the vortex and the edge. Upon entrance of the vortex from the right edge, the edge force slows down the vortex. On the contrary, upon exit, it adds up with the Lorentz force and increases the velocity. The vortex exit can be understood as the annihilation of a vortex–antivortex pair. This is practically an instant event that is limited only by the relaxation time. The seeming velocity upon annihilation, ∼$2\lambda_{L}/\tau_{GL}=1326$ km/s, is indeed comparable to the maximum velocity in the leftmost section, $x_{4}$ in Figure 5e.

Figure 5f shows the mean vortex velocity upon traversal across the entire device length as a function of current. Blue symbols represent calculated values and the red line marks the linear Bardeen–Stephen approximation; see Equation (Equation 4). The vortex does not enter below $I≲0.31\;I_{dp}$, but once entering, it appears in the nonlinear regime. At the highest current, the mean velocity is about four times larger than the linear approximation.

There are several mechanisms of nonlinear vortex viscosity at high propagation velocities [50,51,52]; for a recent review, see, e.g., Ref. [47]. Within the TDGL formalism, the nonlinearity is caused by the nonequilibrium expansion (distortion) of the vortex core. The finite relaxation time, $\tau_{GL}$, should lead to some shrinkage of the core in front and the appearance of a tail with a reduced (unrelaxed) order parameter behind a moving AV [44]. The faster the vortex velocity, the longer the tail. Qualitatively, this leads to the expansion of the effective ξ in Equation (Equation 3), leading to a reduction in viscosity. However, in the considered case, there are several additional factors affecting vortex velocity.

First, from Figure 5c, it can be seen that the order parameter is significantly reduced on the left side of the vortex core. That is, the major expansion of the core occurs in front and not behind the moving vortex. A similar suppression of the order parameter at the edges has been reported in earlier TDGL simulations [39,40]. It could be attributed to AV interaction with an image’s antivortex [49]. Physically, the suppression of ${\left|\psi \right|}^{2}$ is caused by current crowding at the left side, where vortex and transport currents add up; see the current map in Figure 3e. Thus, we can identify two edge effects, enabling ultrafast vortex propagation: the interaction with an image antivortex exerts an additional driving force on the AV and simultaneously reduces the flux-flow viscosity due to the modification (expansion) of the core shape.

Second, in our case, the vortex is moving in the track with a reduced ${\left|\psi \right|}^{2}$. The strong influence of the track width on the vortex velocity can be deduced from Figure 2c. It is seen that $\tau_{01}$ is rapidly decreasing with the track width increasing within $\xi <W<\lambda_{L}$. The suppressed order parameter in the track leads both to the significant expansion of the vortex core, as can be seen from the elongated core shape in Figure 5b, and to the reduction of $I_{dp}^{*}$ in the track. At *I* larger than $I_{dp}^{*}$, the track may start acting as a Josephson junction [53], enabling much faster propagation velocities [22]. From Figure 2c, it follows that the average AV velocity increases by a factor of three with increasing *W*. Thus, the high AV propagation velocities in the considered devices reflect the significant amplification of nonlinear effects in the track.

### 2.3. AVRAM Cell Dynamics

In Figure 3a–c, the current dependencies of the three characteristic time scales are shown for an optimized cell with $z_{n}=0$, $W≃1.06\;\lambda_{L}$ and $D≃1.5\;\lambda_{L}$. Figure 3d–f represent the corresponding color maps of the order parameter and the current density in the cell.

Figure 3a shows the write time, $\tau_{01}$, by a positive current. The inset illustrates that the writing operation in this case is achieved by a guided motion from the notch to the trap along the track. It is seen that $\tau_{01}$ rapidly drops above a threshold write current, which for the chosen set of parameters is $I_{01}≃0.418\;I_{dp}$.

At $I_{01}<I<I_{dp}$, an AV will certainly reach the trap, but it will not necessarily stay there indefinitely. Figure 3b shows the vortex residence time, $\tau_{r}$, as a function of positive current. Above a pinning current, $I_{p}≃0.424\;I_{dp}$ in this case, the vortex escapes from the trap and moves toward the left edge (outside the track), as sketched in the inset. Therefore, the deterministic write operation, independent of the current pulse duration, can be achieved at $I_{01}<I<I_{p}$.

Figure 3c shows the erase time, $\tau_{10}$, by a negative current. In this case, the vortex is guided out via the track, as sketched in the inset. Due to the geometric asymmetry, the threshold erase current, $|I_{10}|≃0.075\;I_{dp}$, is significantly smaller than $I_{01}$. With increasing negative current, $\tau_{10}$ rapidly decreases. However, at a larger negative current, $I_{1\bar{1}}≃-0.391\;I_{dp}$, an antivortex will be written into the trap via the track after erasure of the trapped vortex. Note that $|I_{1\bar{1}}|$ is smaller than $I_{01}$. This can be explained by a nonequilibrium suppression of the order parameter around the area of vortex–antivortex annihilation in the track, which effectively lowers the value of $I_{01}$ for a short relaxation period and allows the entrance and trapping of a subsequent antivortex. The erase operation is deterministic at $I_{1\bar{1}}≲I≲I_{10}$. In the considered case, the erase current range, $\left[-0.075,-0.391\right]\;I_{dp}$, is very broad.

For reliable operation of the cell, the inequality $|I_{10}|$$<I_{01}<I_{p}$ must hold true (implying $|I_{1\bar{1}}|$$<I_{01}$) for the three threshold currents, which is consistent with the experimental observations [24]. This is not given for granted, but it can be achieved with conscious geometric design.

In Figure 4a, we show current dependencies of write (top) and residence (bottom) times for different track widths and $D≃1.5\;\lambda_{L}$. In Figure 2c, it has been shown that $\tau_{01}$ decreases with increasing *W*. From Figure 4a, we can see that the write threshold, $I_{01}$, has a similar tendency. On the other hand, $I_{p}$ and $\tau_{r}$ are practically independent of *W* because a vortex unpinned by a positive current continues its traversal toward the left edge and not backwards along the track.

Figure 4b,c show a correlation between $\tau_{01}\left(I\right)$ and $\tau_{r}\left(I\right)$ for different *W* and *D* in a double-logarithmic scale. The dashed lines mark $\tau_{r}=\tau_{01}$. Generally, long $\tau_{r}$ and short $\tau_{01}$ is desired for memory application. Therefore, cells with behavior corresponding to $\tau_{r}>\tau_{01}$ (above the $\tau_{r}=\tau_{01}$ line) allow for a higher controllability of vortex dynamics, and such cells would be more tolerant to the variations in the write current pulse duration. For *W* and *D* comparable to $\lambda_{L}$, there exists a range of currents facilitating fully deterministic write operation with $\tau_{r}=\infty$, as follows from Figure 3a,b.

### 2.4. Flux-Flow State

Figure 6a–c show the time evolution of the flux in the trap after the application of current for an optimized AVRAM cell with $D=1.5\lambda_{L}$, $W≃\lambda_{L}$, and $z_{n}=0$. Figure 6a corresponds to $I/I_{dp}≃0.42$. In this case, a vortex enters from the notch at the right edge, travels along the track, reaches the trap at $\tau_{01}≃182$ ps and becomes trapped indefinitely, $\tau_{r}=\infty$. This is the ideal deterministic write operation, which is independent of the bias time, $t_{b}$, provided $t_{b}\geq \tau_{01}$.

Figure 6b represents the time evolution at a slightly larger current, $I/I_{dp}=0.45$. The AV initially reaches the trap after $\tau_{01}≃19$ ps, stays there for a short time and subsequently leaves the device through the left edge. The process is repeated with the period ∼37 ps. Thus, the device enters the flux-flow state with a periodic, one-by-one entrance and exit of AV.

A further increase of current leads to faster dynamics with a shortening of all the time scales. Noticeably, at higher currents, the flux flow becomes aperiodic with subsequent vortices moving faster than the previous, as seen in Figure 6c. This is clear evidence of the nonequilibrium reduction of flux-flow viscosity, which was discussed earlier. It becomes pronounced at high currents and velocities when the vortex time of flight becomes shorter than 10 ps (∼$37\;\tau_{GL}$). This leads to a formation of a phase-slip line [53] with a reduced unrelaxed order parameter across the entire device length. It acts as a self-established guiding track [54], enabling ultrafast vortex motion due to the reduced viscosity.

## 3. Discussion

The key problem of SQUID-based RSFQ electronics is the lack of scalability. To store $\Phi_{0}$, a SQUID should have a parameter $\beta_{L}=2LI_{c}/\Phi_{0}>1$, where *L* is the SQUID loop inductance and $I_{c}$ is the Josephson critical current. Consequently, the inductance should be larger than $L>\Phi_{0}/2I_{c}$. Upon the miniaturization of Josephson junctions, $I_{c}$ decreases and the loop, on the contrary, has to be made larger. Together with a complex architecture and the need for an additional readout SQUID, this limits the size of the RSFQ memory cell to ∼10μm [14], making it incompatible with VLSI [12].

AV-based electronics facilitate a miniaturization down to sub-micron sizes [24] because $\Phi_{0}$ is stored by the means of pinning without the need for large inductance. At the same time, since the AV carries the flux quanta, vortex-based electronics may allow for the utilization of the key aspects from the RSFQ ideology—namely operation with quantized RSFQ pulses.

Geometry is playing a crucial role in quantum dots. We confirm the conclusion of Ref. [24] that a specific geometrical asymmetry is required for deterministically controllable vortex manipulation. In our case, the asymmetry is achieved with the help of a notch for vortex entrance and a track for guiding the AV into the trap. We have shown that the choice of track width and trap diameter are of high significance. According to our analysis, the optimal size for both elements is determined by $\lambda_{L}$. We note that the number of relevant geometrical parameters is very large, and other types of asymmetries can also be utilized. Geometrical confinement effects lead to the appearance of preferable vortex positions in a fluxonic quantum dot [32,33,34,35]. Therefore, the shape of the island and the position and shape of the trap will affect device operation.

### 3.1. Manipulation by Short Current Pulses

The AV in AVRAM cells is manipulated by current pulses [23,24]. The AV dynamics is controlled by three threshold currents:

$I_{01}$—the positive write current along the track;

$I_{p}$—the positive de-pinning current in the direction opposite to the track;

$I_{10}$—the negative current for erasing along the track.

The deterministic operation is achieved at the condition
(6)$$|I_{10}|<I_{01}<I_{p}.$$
The first inequality prevents the trapping of an antivortex after erasing the vortex. The second inequality prevents vortex escape during the write operation. If the conditions of Equation (Equation 6) are satisfied, there exist ranges of current in which the cell operation is insensitive to pulse duration (provided it is longer than the corresponding times of flight $\tau_{01}$ or $\tau_{10}$). The deterministic write operation is achieved for $I_{01}<I<I_{p}$, with $I_{01}≃0.418\;I_{dp}$ and $I_{p}≃0.424\;I_{dp}$, as can be seen from Figure 4a. The deterministic erase operation occurs at $I_{1\bar{1}}<I<I_{10}$ with $I_{1\bar{1}}≃-0.391\;I_{dp}$ and $I_{10}≃-0.075\;I_{dp}$ for the given choice of cell parameters.

The AVRAM operation times strongly depend on the current amplitude. Figure 7a,b show the dynamics of (a) 0→1 switching at $I≃0.419\;I_{dp}$ and (b) 1→0 switching at $I≃-0.08\;I_{dp}$. These currents are only slightly above the corresponding threshold currents, leading to relatively long switching times, $\tau_{01}≃100$ ps and $\tau_{10}≃43$ ps. Figure 7c demonstrates a series of repeated write/erase operations by such currents with pulse lengths $t_{01}=104$ ps and $t_{01}=44$ ps.

### 3.2. Stroboscopic Effect at Large Currents

The operation speed of AVRAM cells is limited by the AV time of flight, which strongly depends on current, as can be seen from Figure 3, Figure 4, Figure 5 and Figure 6. Larger currents allow faster operation. As seen in Figure 6c, the time of flight through an $L_{x}=1\;\mu$m cell can be as low as a few ps, and it would be even shorter for smaller cells. Thus, larger currents can enable ultrafast switching. However, this comes at the expense of switching stability. As shown in Figure 6b,c, at $I>I_{p}$, the device enters into the flux-flow state. In this case, the final state of the device becomes dependent on the pulse length. If the time of the pulse, $t_{p}$, is much longer than $\tau_{01}+\tau_{r}$, the cell will exhibit a stroboscopic effect, as reported in experiment [24], with the final state depending on the ratio $t_{p}/\left(\tau_{01}+\tau_{r}\right)$.

Figure 7d,e show switching dynamics at larger currents (d) $I≃0.478\;I_{dp}>I_{p}$ and (e) I≃−0.39$I_{dp}≃I_{1\bar{1}}$. The corresponding switching times are $\tau_{01}≃10.5$ ps and $\tau_{10}≃2.6$ ps, which are approximately an order of magnitude shorter than in Figure 7a,b. Figure 7f demonstrates a controllable write/erase operation by such currents with short pulse lengths $t_{01}=13$ ps and $t_{01}=4$ ps. This indicates that AVRAM is capable of operation at frequencies in the range of 100 GHz. Thus, it is possible to enable fast vortex manipulation by high-amplitude current pulses. However, this requires the ps control of pulse lengths.

### 3.3. Estimations for Nb-Based Cells

The AVRAM cell, studied in Ref. [24], Figure 1, is made of sputtered Nb film. According to Equation (Equation 4), the switching speed of the AVRAM cell depends on $\sigma_{n}$ and ξ. While the used value of $\sigma_{n}$ is realistic for Nb films, the adopted value of ξ is approximately three times larger than the $\xi_{0}≃14$ nm of sputtered Nb films [55] in order to simplify (increase the mesh size) and speed up the calculations. Therefore, in order to make estimations for Nb films, we should properly scale the obtained values. For Nb, the value of flux-flow viscosity, $\eta \propto \xi^{-2}$ Equation (Equation 3), will be an order of magnitude larger, and flux-flow velocities, Equation (Equation 4), will be an order of magnitude smaller.

So far, AVs were considered as very slow objects with maximum velocities in the range of ∼1 km/s [44,45,46,47], which would make them unsuitable for high-frequency operation. Here, we have demonstrated a possibility of ultrafast vortex motion with velocities up to ∼1000 km/s; see Figure 5. In case of Nb films, it would be an order of magnitude smaller, ∼100 km/s, which is still much larger than the velocities of a few km/s reported for macroscopic films [47]. This is the most important new result of this work.

We have identified several mechanisms enabling ultrafast vortex motion.

(i)Edge effect in a mesoscopic superconductor. We consider vortex motion in a fluxonic quantum dot of dimensions comparable to the vortex size. This leads to the appearance of strong edge forces, acting on the vortex. The attractive edge–vortex force can be considered as due to the vortex–image antivortex interaction [49]. At small distances, the effective edge current density acting on the vortex approaches $J_{dp}$. According to Equation (Equation 1), this enables the maximum possible driving force, which is not achievable by the bias current.(ii)The spatial asymmetry of the cell. In a spatially symmetric superconductor, edge forces are equal in amplitude and opposite in direction at the two edges. Therefore, they will slow down the AV at the entrance and speed up at the exit. Although the net effect is nonzero because of nonlinear viscosity at high speed, it will be small due to the mutual cancellation of forces. Here, we consider a spatially asymmetric cell. The presence of a notch and a track reduces the image force at the AV entrance and, thus, removes edge force cancellation, increasing the net velocity.(iii)The guiding track. Write and erase operations are achieved via a track with a reduced superconducting order parameter. As seen from Figure 2c, a track with *W*∼$\lambda_{L}$ enables more than a three-fold increase in the velocity both due to the reduction of viscosity caused by nonequilibrium core expansion and by the enhancement of flux-flow nonlinearity due to the reduction of $I_{dp}^{*}$ and thus enhancement of $I/I_{dp}^{*}$ in the track. The maximum AV velocity is determined by the track depth, i.e., by the suppression of the order parameter. Here, we presented data for a modest track depth, $|\psi {|}^{2}$=0.6. Any further reduction of $|\psi {|}^{2}$ would lead to faster vortex motion. However, at $|\psi {|}^{2}$≪1, the track would become a Josephson junction, consequently turning the Abrikosov vortex into a Josephson vortex. Although the Josephson vortex can propagate at the speed of light (in the transmission line), it is difficult to pin and store [22]. Furthermore, in this case, the cell would become equivalent to the RF-SQUID and would need a significant trap-hole inductance for storing $\Phi_{0}$, causing the same problem with miniaturization as for RSFQ memory.

### 3.4. Perspectives of AVRAM

Finally, we discuss the perspectives and limitations of AVRAM.

**Size**. The miniaturization of cells to the submicron size is required for reaching the integration level typical for contemporary semiconducting electronics. A prototype of a 1 × 1 μm^2^ AVRAM cell, operational at zero magnetic field, has already been demonstrated [24]; see Figure 1. The limit of miniaturization is determined by the range of stability of the vortex in the trap. At H=0, the vortex is metastable; i.e., it has a surplus of energy. The reason why it does not disappear is the pinning on the trap. The hole reduces vortex energy and thus creates a potential well, preventing spontaneous vortex escape. On the other hand, edge forces (image antivortex) try to pull it out. From Figure 5d, it can be seen that this force is rapidly increasing when the vortex-edge distance becomes less than $\lambda_{L}$. Thus, it will become increasingly difficult to store a vortex in islands less than $2\lambda_{L}$. However, the theoretical limit of stability is much smaller ∼2ξ because in a symmetric island with a vortex in the middle, edge forces cancel out. However, as we emphasized above, the vortex cannot be written in the symmetric cell. Therefore, the actual limit will depend on subtle effects related to a specific geometry.

Thus, the anticipated limit of island miniaturization is *L*∼$2\lambda_{L}$∼200 nm for Nb. Although the footprint on a chip will be somewhat larger, this could enable the giga-scale integration level compatible with modern semiconducting technology. The competitiveness of vortex-based electronics is further enhanced by its advanced functionality. Indeed, the AVRAM cell is nonvolatile and will hold the information without power supplied to the cell (provided the chip is kept cold). Therefore, performance-wise, it is better than the static SRAM, which for the 7 nm seminconducting technology has the footprint size of ∼230 nm [56].

**Speed**. Our simulations indicate that AVRAM cells enable deterministic (pulse time-insensitive) switching times of 100 ps. Faster switching at time scales of a few ps is achievable at larger current amplitudes but requires the ps control of current pulses. The latter, however, is a prerequisite for ultrafast electronics. The ultrafast operation speed is enabled by ultrafast vortex motion in a specially prepared track. Our estimation for Nb films yields the top velocity ∼100 km/s, which is comparable to recent direct experimental observations [54].

**Energy efficiency**. The total energy per operation is equal to the work completed by the Lorentz force [23],
(7)$$E=\frac{I\Phi_{0}}{2}.$$
Here, the factor 1/2 appears because the AV is transported only through the half of the cell. For a comfortable operation at the current of I=100μA [24], $E≃1.0\times 10^{-19}$ J. Thus, AVRAM is characterized by a very low access energy [26].

## 4. Conclusions

We performed the numerical modeling of AVRAM cell dynamics at zero magnetic field with the aim to determine optimal parameters, clarify the operation principle and establish perspectives and limitations of vortex-based memory. We obtained the following most notable results.

Cell operation requires a specific geometrical asymmetry, which in our case is achieved by adding a notch for the vortex entrance and a track with a reduced-order parameter for guiding the vortex into the trap. The optimal sizes of these components are determined by the London penetration depth, which also sets the limit for miniaturization.

The current ranges and conditions for deterministic vortex manipulation, as shown in Equation (Equation 6), were defined. A controllable write/erase operation with switching times of a few ps and operation frequencies of ∼100 GHz was demonstrated. The ultrafast operation is enabled by an ultrafast vortex dynamics, which is achievable in fluxonic quantum dots with a specific geometry.

We conclude that vortex-based electronics can be employed for building a superconducting digital computer with high speed and energy efficiency and the integration level approaching the existing semiconducting technology.

## Figures and Tables

**Figure 1 nanomaterials-14-01634-f001:**
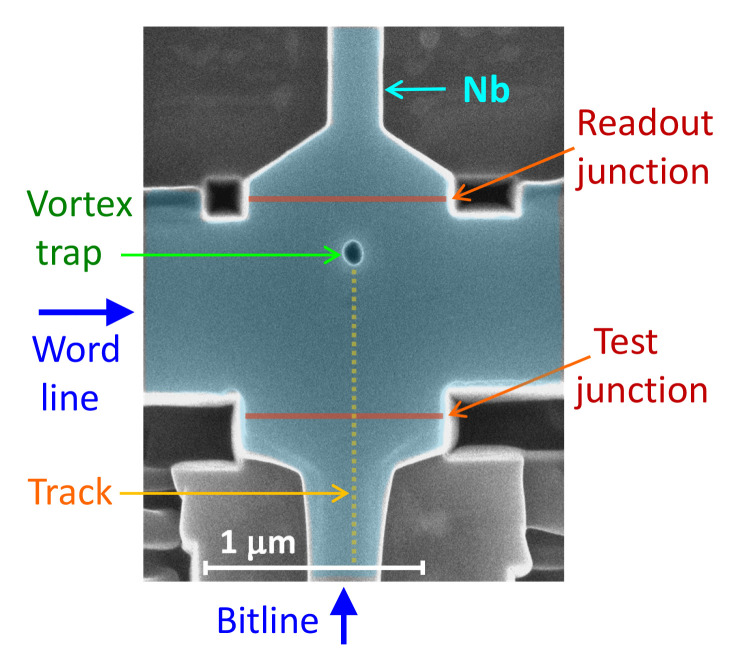
SEM image of an Nb-based AVRAM cell prototype from Ref. [24] The cell contains a superconducting island ∼1 × 1 μm^2^, a vortex trap (a hole in the film), a vortex-guiding track, and two readout Josephson junctions.

**Figure 2 nanomaterials-14-01634-f002:**
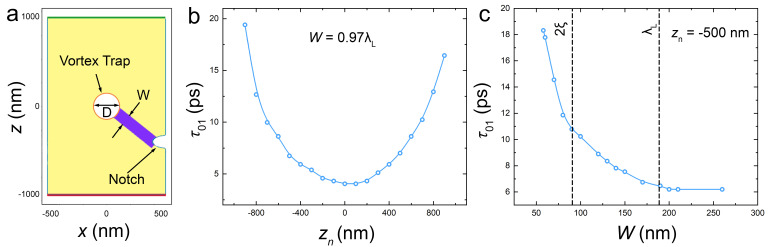
Optimization of the notch and the track. (**a**) Structure of the considered AVRAM cell: a rectangular superconducting film 1 × 2 μm^2^ with a circular vortex trap in the middle with a diameter *D*, a guiding track with a width *W*, and a notch on the right edge and at a vertical position $z_{n}$. Panels (**b**,**c**) show calculated vortex trapping time, $\tau_{01}$, (**b**) as a function of the notch position, $z_{n}$, for a fixed $W=0.97\lambda_{L}$; and (**c**) as a function of the track width, *W*, for $z_{n}=-500\;\mathrm{nm}$. Simulations are made at a constant applied current $I/I_{dp}=0.78$ and $D≃\lambda_{L}$.

**Figure 3 nanomaterials-14-01634-f003:**
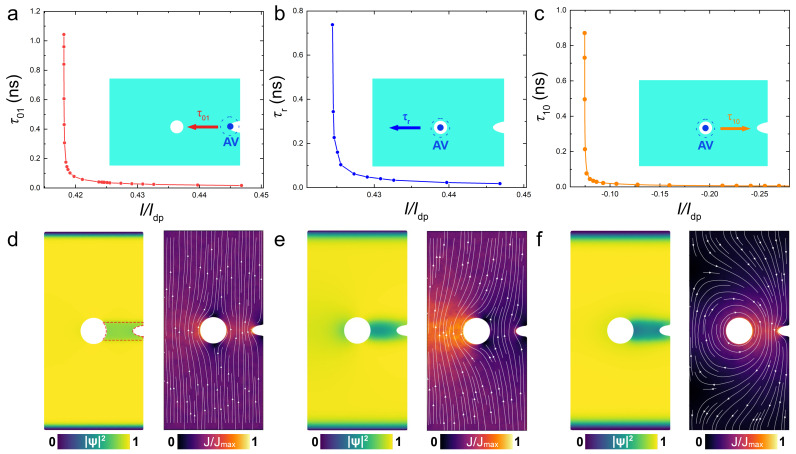
Characteristic times in the optimized cell. (**a**–**c**) Current dependencies of (**a**) write time by a positive current, (**b**) residence time in the presence of a positive current and (**c**) erase time by a negative current. Insets illustrate directions of vortex motion. Panels (**d**–**f**) show color maps of the order parameter (left) and current density (right) in the corresponding cases. (**d**) At positive current without a vortex in the trap. (**e**) At a positive current with a trapped vortex. (**f**) At a negative current with a trapped vortex. Simulations are performed for a cell with $z_{n}=0$, $W≃1.06\;\lambda_{L}$ and $D≃1.5\;\lambda_{L}$.

**Figure 4 nanomaterials-14-01634-f004:**
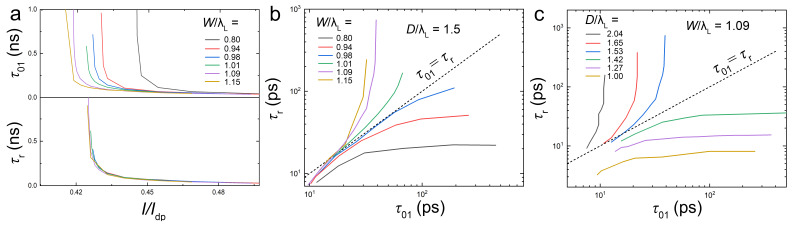
Determination of geometrical parameters for controllable operation. (**a**) Current dependence of the write (top) and residence (bottom) times for different track widths and $D≃1.5\lambda_{L}$. (**b**) A correlation between $\tau_{r}$ and $\tau_{01}$ for the data from (**a**). (**c**) A correlation between $\tau_{r}$ and $\tau_{01}$ for different trap diameters and $W=1.09\;\lambda_{L}$. Dashed lines in (**b**,**c**) correspond to $\tau_{r}=\tau_{01}$. A reliable cell operation can be achieved above these lines.

**Figure 5 nanomaterials-14-01634-f005:**
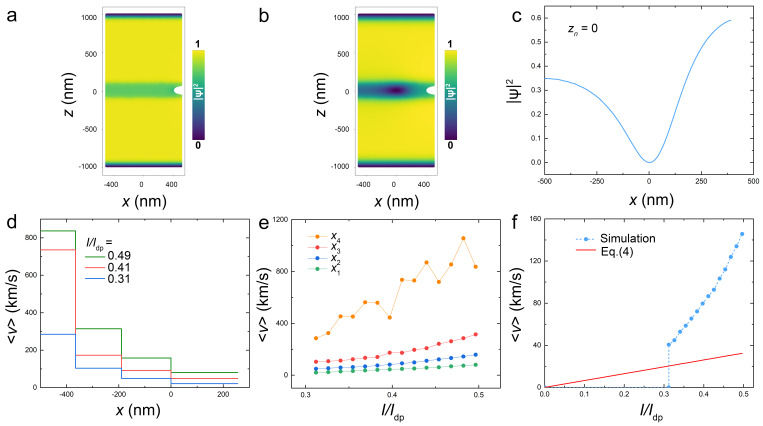
Vortex velocimetry in the track without a trap. (**a**,**b**) Color maps of the order parameter (**a**) without a vortex and (**b**) with a moving vortex from the notch to the left edge at $I/I_{dp}=0.31$. (**c**) A cross-section of the core along the track from (**b**). A significant reduction of the order parameter occurs in front (at the left) of the vortex. (**d**) Average vortex velocities in four sections of the track, $x_{4}=\left[-500,-365.8\right]$ nm, $x_{3}=\left[-365.8,-190\right]$ nm, $x_{2}=\left[-190,0\right]$ nm, $x_{1}=\left[0,255.8\right]$ nm, at three different currents. (**e**) Average velocities in the same sections as a function of current. A strong increase in velocity at the left edge is due to interaction with an image antivortex. (**f**) The net average vortex velocity in the track as a function of current (blue circles). The red line show the linear Bardeen–Stephen approximation.

**Figure 6 nanomaterials-14-01634-f006:**
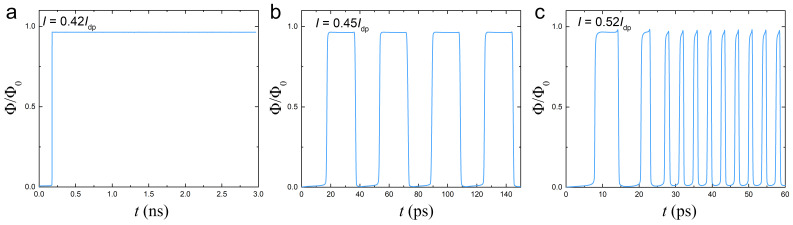
Stroboscopic behavior in the flux-flow state. The time dependence of the flux in the trap after application of constant current. (**a**) At low current, $I_{01}<I≃0.42$$I_{dp}<I_{p}$, the vortex slowly arrives at the trap and stays there indefinitely long despite the applied current. At higher currents, the cell enters in the fast flux-flow state, which is (**b**) periodic at not very large *I* but becomes (**c**) ultrafast and aperiodic at larger *I*. In the flux-flow state, the cell exhibits a stroboscopic effect with respect to the duration of the current pulse.

**Figure 7 nanomaterials-14-01634-f007:**
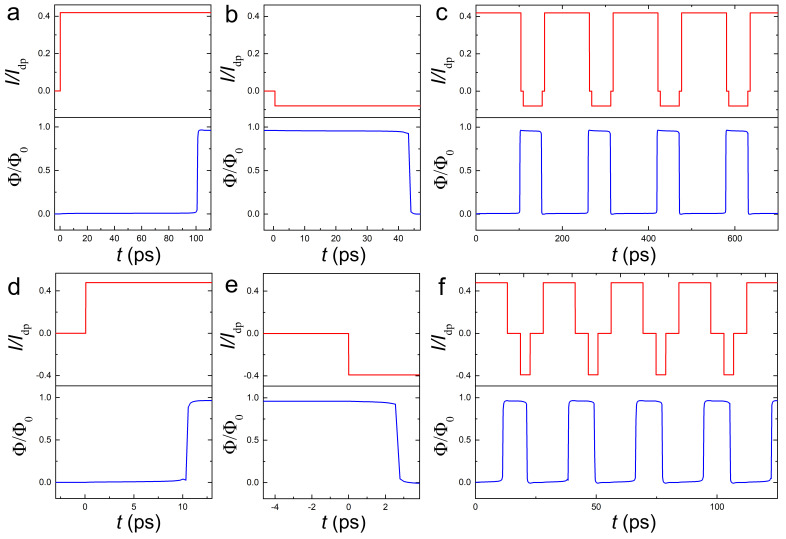
Demonstration of vortex manipulation by current pulses. Deterministic switching between states using positive and negative current pulses, optimized for the chosen geometry to write and erase the vortex from the trap. (**a**) Low current with long switching time. (**b**) High current with short switching time.

## Data Availability

The data presented in this study are available on request from the corresponding author.

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
