# Peer review of "Numerical Modeling of Vortex-Based Superconducting Memory Cells: Dynamics and Geometrical Optimization"

_nanomaterials, 2024, doi:10.3390/nano14201634_

Round 1
Reviewer 1 Report
Comments and Suggestions for Authors
The authors e present numerical modeling 5 of such cells using time-dependent Ginzburg-Landau equations. The cell represents a fluxonic 6 quantum dot containing a small superconducting island, an asymmetric notch for vortex entrance, 7 a guiding track, and a vortex trap. We determine the optimal geometrical parameters for operation 8 at zero magnetic field and the conditions for controllable vortex manipulation by short current 9 pulses. We report ultra-fast vortex motion with velocities more than an order of magnitude faster 10 than those expected for macroscopic superconductors. This phenomenon is attributed to strong 11 interactions with the edges of a mesoscopic island, combined with the nonlinear reduction of 12 flux-flow viscosity due to nonequilibrium effects in the track. Our results show that such cells can 13 be scaled down to sizes comparable to the London penetration depth, ∼ 100 nm, and can enable ultrafast switching on the picosecond scale with ultra-low energy per operation, ∼ 10−19 14 J. This is now much impressive due to the following below reasons. I must reject it.
1. Although the use of time-dependent Ginzburg-Landau calculations to simulate AVRAM cells is intriguing, this study adds nothing new to the body of knowledge on vortex dynamics in superconductors. Since Abrikosov vortex storage has been studied previously, the novelty of this work does not appear to be significant enough to justify publishing.
2. The dynamics of the vortex and its interaction with the mesoscopic island's boundaries, along with the nonlinear reduction of flux-flow viscosity, are intricately described and challenging to understand. Better explanations and a more approachable exposition of the relevant physical ideas would improve the manuscript.
3. The mechanism of brief current pulses for manipulating vortices in a controllable manner is not well understood. More information on the specific mechanisms of manipulation and the consistency and dependability of the process in various cells or under different situations would enhance the work.
Comments on the Quality of English LanguageThe authors e present numerical modeling 5 of such cells using time-dependent Ginzburg-Landau equations. The cell represents a fluxonic 6 quantum dot containing a small superconducting island, an asymmetric notch for vortex entrance, 7 a guiding track, and a vortex trap. We determine the optimal geometrical parameters for operation 8 at zero magnetic field and the conditions for controllable vortex manipulation by short current 9 pulses. We report ultra-fast vortex motion with velocities more than an order of magnitude faster 10 than those expected for macroscopic superconductors. This phenomenon is attributed to strong 11 interactions with the edges of a mesoscopic island, combined with the nonlinear reduction of 12 flux-flow viscosity due to nonequilibrium effects in the track. Our results show that such cells can 13 be scaled down to sizes comparable to the London penetration depth, ∼ 100 nm, and can enable ultrafast switching on the picosecond scale with ultra-low energy per operation, ∼ 10−19 14 J. This is now much impressive due to the following below reasons. I must reject it.
1. Although the use of time-dependent Ginzburg-Landau calculations to simulate AVRAM cells is intriguing, this study adds nothing new to the body of knowledge on vortex dynamics in superconductors. Since Abrikosov vortex storage has been studied previously, the novelty of this work does not appear to be significant enough to justify publishing.
2. The dynamics of the vortex and its interaction with the mesoscopic island's boundaries, along with the nonlinear reduction of flux-flow viscosity, are intricately described and challenging to understand. Better explanations and a more approachable exposition of the relevant physical ideas would improve the manuscript.
3. The mechanism of brief current pulses for manipulating vortices in a controllable manner is not well understood. More information on the specific mechanisms of manipulation and the consistency and dependability of the process in various cells or under different situations would enhance the work.
Author Response
Reviewer#1 writes:
“1. Although the use of time-dependent Ginzburg-Landau calculations to simulate AVRAM cells is intriguing, this study adds nothing new to the body of knowledge on vortex dynamics in superconductors. Since Abrikosov vortex storage has been studied previously, the novelty of this work does not appear to be significant enough to justify publishing.”
Reply-1
We agree that it is important to clarify what has been already done and what is new. There has been a large volume of earlier numerical studies within TDGL formalism. We cite some of them and apologize for not being able to include all. The concept of AV-memory was also already demonstrated experimentally by our group. Experimentally, we have observed that subtle geometrical factors can play a significant role. It was not clear for us why this happens and how we can control and utilize it. Therefore, we performed numerical modeling of the particular cell (with specific dimensions and geometry) in order to understand the intricate cell dynamics. Our goal was to optimize the next generation of devices. Modelling was the cheapest and fastest way to do that. We did not plan a publication, just wanted to understand how it works. However, some of the results appeared to be quite surprising and, to our knowledge, new. For example, we figured out that the optimal size of the trap and track is determined by \lambda, not \xi, which was not obvious to us. But, most remarkably, we observed an ultrafast vortex propagation in the track, one-two orders of magnitude faster than anticipated in a uniform superconductor. This is an illustration of a subtle feature having a huge effect. Knowing this is important for designing such devices. We believe that vortex-based electronics has a future and want to share this information with the scientific community so that such electronics could be reproduced and further developed by others. This is our motivation for publication.
Reviewer #1 writes:
“2. The dynamics of the vortex and its interaction with the mesoscopic island's boundaries, along with the nonlinear reduction of flux-flow viscosity, are intricately described and challenging to understand. Better explanations and a more approachable exposition of the relevant physical ideas would improve the manuscript.”
Reply-2
In reply to this critics, in the modified version we extended the discussion and provided additional clarifications of underlying physics.
Reviewer #1 writes:
“3. The mechanism of brief current pulses for manipulating vortices in a controllable manner is not well understood. More information on the specific mechanisms of manipulation and the consistency and dependability of the process in various cells or under different situations would enhance the work.”
Reply-3
We agree with the Reviewer. Corresponding changes are introduced in the modified version.
Reviewer 2 Report
Comments and Suggestions for Authors
The paper is devoted to theoretical modeling of an asymmetric superconducting cell with a vortex trap and optimization of its geometry. The paper is written in good language and I have no questions about the form of presentation of the material. However, there are several significant comments on the meaning of the work.
- The authors write about Abrikosov vortices in a zero magnetic field. It is known that an Abrikosov vortex is a current vortex around quasi-one-dimensional magnetic flux threads (of 1 flux quantum magnitude) that are formed in a type II superconductor when it is placed in an external magnetic field. When the field is reduced to 0, the vortices disappear. In this regard, a question arises about the nature of the objects under consideration - what is an Abrikosov vortex in a zero external field? Or are we talking about Josephson vortices, the total magnetic flux of which is 0?
- The authors rather carelessly use the terms "quantum dot" and "mesoscopic island" in relation to the same object. It seems to me that the term "mesoscopic island" is closer to the truth, since "quantum dot" implies the presence of quantum confinement effects and 3D localization of the object (vortex) inside, without the possibility of free movement.
- The authors claim complete non-volatile character of the memory cell based on the discussed vortices (and low switching energy, at the level of 1e-19 J). This correct within the framework of the low-temperature regime (5-6 K, as indicated in the authors' previous works), which is necessary to maintain the superconducting state of the substance. It is known that cryogenic systems consume a significant energy. In this regard, discussions about the non-volatile memory seem somewhat hasty... At least until the discovery and implementation of high-temperature superconductivity.
- The authors claim that it is possible to miniaturize memory cells down to ~200 nm, presenting this as an advantage and a prospect for obtaining gigabit-sized memory cards. Here I categorically disagree with the authors. It is known that other, more developed types of fast non-volatile memory, such as FRAM and MRAM, have reached a cell size of about 100 nm, and this significantly complicates the manufacture of gigabit memory cards. Taking into account the still cryogenic nature of the discussed AVRAM elements, such a prospect looks very far.
- The authors' calculations are mainly devoted to a large-scale cell (about 1x1 µm). It is clear why such an object was chosen - the authors have experimental data from earlier publications to rely on. However, it is not very clear why the authors could not consider at least a reduced cell (up to ~ 100 nm) within the framework of theoretical calculations? This would allow assessing the prospects for miniaturization.
Author Response
Reviewer #2 writes:
“The paper is devoted to theoretical modeling of an asymmetric superconducting cell with a vortex trap and optimization of its geometry. The paper is written in good language and I have no questions about the form of presentation of the material. However, there are several significant comments on the meaning of the work.”
Reply:
We are grateful to the Reviewer for valuable remarks and constructive critics. We made corresponding clarification and modifications in the resubmitted version.
Reviewer #2 writes:
“1. The authors write about Abrikosov vortices in a zero magnetic field. It is known that an Abrikosov vortex is a current vortex around quasi-one-dimensional magnetic flux threads (of 1 flux quantum magnitude) that are formed in a type II superconductor when it is placed in an external magnetic field. When the field is reduced to 0, the vortices disappear. In this regard, a question arises about the nature of the objects under consideration - what is an Abrikosov vortex in a zero external field? Or are we talking about Josephson vortices, the total magnetic flux of which is 0?”
Reply-1
The vortex structure does change in the mesoscopic island and the net flux decreases below \Phi_0. However, the cell considered here is ~6 times larger \lambda. Therefore, such effects are relatively small and the vortex is more-or-less conventional and carries \Phi_0 flux. The reason why it does not disappear is the pinning on the trap. The hole reduces vortex energy and thus creates a potential well, preventing spontaneous vortex escape. The clarification is added in the modified version.
Reviewer #2 writes:
“2. The authors rather carelessly use the terms "quantum dot" and "mesoscopic island" in relation to the same object. It seems to me that the term "mesoscopic island" is closer to the truth, since "quantum dot" implies the presence of quantum confinement effects and 3D localization of the object (vortex) inside, without the possibility of free movement.”
Reply-2
We fully agree that “quantum dot” is characterized by quantum confinement, which occurs when the dot size becomes comparable to the wavelength. We are talking about a fluxonic quantum dot, in which the confinement is determined by the current distribution scale \lambda. The width of considered cell is ~6 times \lambda and the vortex is placed in the middle, only ~3\lambda away from the edge. This does cause a strong interaction with the edge, as seen from the distortion of vortex shape in Fig. 3 (c). Similarly, in Fig. 3 (d) an increase of vortex velocity is seen at negative x (the vortex trap is at x=0), which occurs because of the confinement (edge) effects. According to Wikipedia, “mesoscopic physics” implies the necessity of quantum-mechanical description, including quantum confinement effects (e.g. conductance quantization). Therefore, to our opinion, “mesoscopic island” and “quantum dot” are synonyms and interchangeable usage of these terms may be confusing, but not erroneous. We want to retain the “fluxonic quantum dot” term because it is colloquially known and informative.
Reviewer #2 writes
“3. The authors claim complete non-volatile character of the memory cell based on the discussed vortices (and low switching energy, at the level of 1e-19 J). This correct within the framework of the low-temperature regime (5-6 K, as indicated in the authors' previous works), which is necessary to maintain the superconducting state of the substance. It is known that cryogenic systems consume a significant energy. In this regard, discussions about the non-volatile memory seem somewhat hasty... At least until the discovery and implementation of high-temperature superconductivity.”
Reply-3:
For cryo-electronics we have to count separately power at and outside the chip. The power budget at a chip is very small (~1W). Therefore, a very small access energy, < 1 aW, is required. The refrigeration power outside the chip is many orders of magnitude larger, but does not directly affect the operation. In the modified version we added a clarification that AVRAM in non-volatility in a sense that it does not need to be powered, but it still needs to be cooled.
Reviewer #2 writes
“4. The authors claim that it is possible to miniaturize memory cells down to ~200 nm, presenting this as an advantage and a prospect for obtaining gigabit-sized memory cards. Here I categorically disagree with the authors. It is known that other, more developed types of fast non-volatile memory, such as FRAM and MRAM, have reached a cell size of about 100 nm, and this significantly complicates the manufacture of gigabit memory cards. Taking into account the still cryogenic nature of the discussed AVRAM elements, such a prospect looks very far.”
Reply-4
We do agree. The 200nm ~2\lambda is the ultimate size of the island. The foot-print of the cell will be somewhat larger. The corresponding statement is added
Reviewer #2 writes:
“5. The authors' calculations are mainly devoted to a large-scale cell (about 1x1 µm). It is clear why such an object was chosen - the authors have experimental data from earlier publications to rely on. However, it is not very clear why the authors could not consider at least a reduced cell (up to ~ 100 nm) within the framework of theoretical calculations? This would allow assessing the prospects for miniaturization.”
Reply-5
Our primary goal was to do numerical modeling of the existing 1x1um^2 cell. This determines sizes and geometries. Limits of miniaturization are indeed very important. But, our simulations do provide a clear answer on this question. As we noted in Reply-1, the stability of vortex is determined by vortex pinning in the trap. The force that pushes it out is the edge force from the image antivortex. From
Fig. 3 (d) it can be seen that this force is rapidly increasing when the vortex-edge distance becomes less than \lambda. Thus, it will become increasingly difficult to store a vortex in islands less than 2\lambda – which leads to our estimation of the ultimate size. However, the theoretical limit of stability is much smaller ~2\xi because in a symmetric island with a vortex in the middle, edge forces cancel out. However, as we emphasized in the manuscript, vortex cannot be written in the symmetric cell. Therefore, the actual limit will depend on subtle effects related to a specific geometry. We added corresponding clarification in the modified version.
Reviewer 3 Report
Comments and Suggestions for Authors
This theoretical paper is a continuation of a series of works by one of the authors on the topic of application of the Abrikosov vortex as a fundamental element for superconducting electronics (superconducting memory). The phenomenological approach in the form of the formalism of the extended nonstationary Ginzburg-Landau equations was employed for the simulation. The paper is clearly written and coherently presented. However, before making my decision I would like to clarify a few important details about how these results were obtained and interpreted.
1) First of all, I am concerned that Appendix A does not contain boundary conditions that are known to unequivocally affect the behavior in such a system described by nonlinear GL equations.
2) Also some important technical details are missing, in particular the meshgrid for the calculations.
3) Since the authors mention the position of the notch, I would recommend using their introduced notation z_n rather than z in Figure 2b. Or do the authors put something different in the definition of z? In this case, does the shape of the notch play any role in carrying out the calculations? In the process of solving equations it has exactly this semi-ellipsoidal shape?
4) Question concerning manipulation with short current pulses. How was the shape of the pulse technically taken into account in the process of solving the GL equations? Was the current-driven or voltage-driven regime with a certain frequency considered? I have this question because the description to Figure 7 is limited to only pictures (a) and (b).
5) The authors mention the occurrence of phase-slip phenomena. I have a question concerning the possible occurrence of chaotic behavior of the order parameter in such a system, which was investigated in a number of papers
(i) https://doi.org/10.1103/PhysRevB.44.875
ii) https://doi.org/10.1103/PhysRevB.84.094527
iii) https://doi.org/10.1063/1.4791774
iv) https://doi.org/10.1103/PhysRevB.96.064507
I recommend the authors to discuss the effect of this phenomenon on the stability of this type of superconducting memory.
Author Response
Reviewer #3 writes:
”This theoretical paper is a continuation of a series of works by one of the authors on the topic of application of the Abrikosov vortex as a fundamental element for superconducting electronics (superconducting memory). The phenomenological approach in the form of the formalism of the extended nonstationary Ginzburg-Landau equations was employed for the simulation. The paper is clearly written and coherently presented. However, before making my decision I would like to clarify a few important details about how these results were obtained and interpreted.”
Reply:
Thank you for encouraging comments and valuable remarks that helped to improve the manuscript.
Reviewer #3 writes:
“1) First of all, I am concerned that Appendix A does not contain boundary conditions that are known to unequivocally affect the behavior in such a system described by nonlinear GL equations.”
Reply-1
We added clarifications about boundary conditions. The device is current biased: boundary conditions assume that bias current with constant density is passing through bottom and top edges of the device.
Reviewer #3 writes:
“2) Also some important technical details are missing, in particular the meshgrid for the calculations.”
Reply -2
The new picture with mesh is added in the modified version.
Reviewer #3 writes:
“3) Since the authors mention the position of the notch, I would recommend using their introduced notation z_n rather than z in Figure 2b. Or do the authors put something different in the definition of z? In this case, does the shape of the notch play any role in carrying out the calculations? In the process of solving equations it has exactly this semi-ellipsoidal shape?”
Reply-3:
Thank you for noting. It should be z_n. The inconsistency is corrected.
Concerning the shape of the notch: yes the size and shape of the notch do have a significant influence (similar to the track). But here we restricted our analysis to the fixed shape, partly to reduce the number of free parameters, and partly because it has already been studied [41].
Reviewer #3 writes:
“4) Question concerning manipulation with short current pulses. How was the shape of the pulse technically taken into account in the process of solving the GL equations? Was the current-driven or voltage-driven regime with a certain frequency considered? I have this question because the description to Figure 7 is limited to only pictures (a) and (b).”
Reply-4:
The cell is current biased. The pulse is generated by instantly changing boundary conditions. We added clarifications in the modified version.
Reviewer #3 writes:
“5) The authors mention the occurrence of phase-slip phenomena. I have a question concerning the possible occurrence of chaotic behavior of the order parameter in such a system, which was investigated in a number of papers
(i) https://doi.org/10.1103/PhysRevB.44.875
- ii) https://doi.org/10.1103/PhysRevB.84.094527
iii) https://doi.org/10.1063/1.4791774
- iv) https://doi.org/10.1103/PhysRevB.96.064507
I recommend the authors to discuss the effect of this phenomenon on the stability of this type of superconducting memory.”
Reply-5:
Thank you for pointing out this issue. We did not observe a chaotic behavior, partly because the cell is current biased (not voltage biased as in most of the mentioned works). However, mostly because of the notch and track, which have a major stabilizing role. Even in a strongly non-equilibrium state, when the phase-slip-line is formed in the track, the rest of the film remains unaffected. Applied current is never large enough to form additional phase-slip-lines in the rest of the film. We added a clarification along with suggested references.
Round 2
Reviewer 1 Report
Comments and Suggestions for Authors
accepted
Reviewer 2 Report
Comments and Suggestions for Authors
Authors give good responce for all comments. The paper can be published.
Reviewer 3 Report
Comments and Suggestions for Authors
I am grateful to the authors for their comprehensive answers to my questions and comments. I certainly recommend this article for publication.